# Development of Novel ^111^-In-Labelled DOTA Urotensin II Analogues for Targeting the UT Receptor Overexpressed in Solid Tumours

**DOI:** 10.3390/biom10030471

**Published:** 2020-03-19

**Authors:** Benjamin Poret, Laurence Desrues, Marc-André Bonin, Martin Pedard, Martine Dubois, Richard Leduc, Romain Modzelewski, Pierre Decazes, Fabrice Morin, Pierre Vera, Hélène Castel, Pierre Bohn, Pierrick Gandolfo

**Affiliations:** 1Institute for Research and Innovation in Biomedicine (IRIB), University of Rouen Normandy, INSERM U1239, DC2N, 76000 Rouen, France; benjamin.poret@gmail.com (B.P.); laurence.desrues@univ-rouen.fr (L.D.); martin.pedard@univ-rouen.fr (M.P.); martine.dubois@univ-rouen.fr (M.D.); fabrice.morin@univ-rouen.fr (F.M.); pierrick.gandolfo@univ-rouen.fr (P.G.); 2EA 4108, Laboratory of Computer Science, Information Processing and Systems (LITIS), team “QuantIF”, Centre Henri Becquerel, 76000 Rouen, France; romain.modzelewski@chb.unicancer.fr (R.M.); pierre.decazes@chb.unicancer.fr (P.D.); pierre.vera@chb.unicancer.fr (P.V.); pierre.bohn@chb.unicancer.fr (P.B.); 3Department of Physiology & Pharmacology, Institute of Sherbrooke, Faculty of Medicine and Health Sciences, Sherbrooke University, Sherbrooke, QC J1H 5N4, Canada; Marc-Andre.Bonin@USherbrooke.ca (M.-A.B.); Richard.Leduc@USherbrooke.ca (R.L.); 4Institute for Research and Innovation in Biomedicine (IRIB), 76000 Rouen, France

**Keywords:** ^111^Indium-DOTA-peptide analogues, urotensin II, UT receptor, carcinoma, radiolabelling

## Abstract

Overexpression of G protein-coupled receptors (GPCRs) in tumours is widely used to develop GPCR-targeting radioligands for solid tumour imaging in the context of diagnosis and even treatment. The human vasoactive neuropeptide urotensin II (hUII), which shares structural analogies with somatostatin, interacts with a single high affinity GPCR named UT. High expression of UT has been reported in several types of human solid tumours from lung, gut, prostate, or breast, suggesting that UT is a valuable novel target to design radiolabelled hUII analogues for cancer diagnosis. In this study, two original urotensinergic analogues were first conjugated to a DOTA chelator via an aminohexanoic acid (Ahx) hydrocarbon linker and then -hUII and DOTA-urantide, complexed to the radioactive metal indium isotope to successfully lead to radiolabelled DOTA-Ahx-hUII and DOTA-Ahx-urantide. The ^111^In-DOTA-hUII in human plasma revealed that only 30% of the radioligand was degraded after a 3-h period. DOTA-hUII and DOTA-urantide exhibited similar binding affinities as native peptides and relayed calcium mobilization in HEK293 cells expressing recombinant human UT. DOTA-hUII, not DOTA-urantide, was able to promote UT internalization in UT-expressing HEK293 cells, thus indicating that radiolabelled ^111^In-DOTA-hUII would allow sufficient retention of radioactivity within tumour cells or radiolabelled DOTA-urantide may lead to a persistent binding on UT at the plasma membrane. The potential of these radioligands as candidates to target UT was investigated in adenocarcinoma. We showed that hUII stimulated the migration and proliferation of both human lung A549 and colorectal DLD-1 adenocarcinoma cell lines endogenously expressing UT. In vivo intravenous injection of ^111^In-DOTA-hUII in C57BL/6 mice revealed modest organ signals, with important retention in kidney. ^111^In-DOTA-hUII or ^111^In-DOTA-urantide were also injected in nude mice bearing heterotopic xenografts of lung A549 cells or colorectal DLD-1 cells both expressing UT. The observed significant renal uptake and low tumour/muscle ratio (around 2.5) suggest fast tracer clearance from the organism. Together, DOTA-hUII and DOTA-urantide were successfully radiolabelled with ^111^Indium, the first one functioning as a UT agonist and the second one as a UT-biased ligand/antagonist. To allow tumour-specific targeting and prolong body distribution in preclinical models bearing some solid tumours, these radiolabelled urotensinergic analogues should be optimized for being used as potential molecular tools for diagnosis imaging or even treatment tools.

## 1. Introduction

For several years, natural or synthetic derivatives radiolabelled ligands for G protein-coupled receptors (GPCRs) have acquired a great clinical interest, mainly in nuclear medicine, for the diagnosis and treatment of some cancers such as neuroendocrine ones [1,2]. GPCRs overexpressed in tumour cells constitute molecular targets of choice in the field of oncology, since agonist ligands classically induce internalization of the ligand-receptor complex allowing sufficient retention of radioactivity within tumour cells to detect them and/or likely to evoke their death [3,4,5]. The other fundamental component for the development of GPCR radioligands for cancer diagnosis is the combination of commonly low expression and limited density of the targeted GPCR in most organs and the concurrent overexpression and high density of these receptors in tumour cells [2,6,7]. The factors that influence the pharmacokinetic characteristics of a radioligand are still not entirely known. Key characteristics of good radioligands are a low molecular weight and well characterised structure, a flexibility to molecular change, a fast and standardised chemical synthesis, tolerance to stringent radiolabelling, favourable pharmacokinetic and high selectivity for the molecular target and relative rapid clearing from the body [8]. To date, one of the most widely used radioligands in nuclear medicine is ^111^In-DTPA-OC (OC, octreotide), which binds the sst2 receptor, overexpressed in neuroendocrine tumours [9,10,11,12,13]. Other radiotracers exploitable in SPECT (single-photon emission computed tomography) have been developed such as ^99^mTc-MIP1404 [14] or ^123^I-MIP1072 [15] which target the prostate-specific membrane antigen, an overexpressed transmembrane protein exploitable for diagnosis [16]. Thus, the use of radiotracers was extended to tumours that overexpress GPCRs, with the development of, for example, ^99^mTc-HABN or ^99^mTc-BBN both targeting Gastrin-releasing peptide receptor with high density in prostate tumour cells [17] and in breast cancer [18].

The most commonly used post-conjugation method involves grafting the chelator onto the ligand and then incorporation of the radioisotope with the chelator [19]. To date, no single radioisotope nor chelators meet all the expected requirements for the diagnosis and treatment of all cancers. Several studies have shown the thermodynamic superiority of new cyclic chelators such as DOTA or NOTA, thus improving the performance of radioligands grafted onto these chelators [20,21]. In particular, theranostic compounds take advantage of the versatility of new chelators such as DOTA to complex radioisotopes capable of emitting γ, β+ or β− [22,23,24], with a very good overall response in patients with sst-positive tumours [25]. Thus, the development of new radiopharmaceutical tools targeting solid tumours not exclusively from neuroendocrine origin, and their refinement according to patients and pathologies, should lead to the improvement of cancer diagnosis and treatment efficacy with reduced side effects.

Urotensin II (UII) is a conserved cyclic peptide of 11 amino-acid in human, originally isolated from the urophysis of a teleostean fish, which shares sequence homologies with somatostatin [26]. This neuropeptide interacts with a class A GPCR named UT, classically coupled to the PLC/IP3/Ca^2+^ pathway [27,28,29]. UII and its receptor are broadly expressed in peripheral organs and especially in the cardiovascular system [30]. UII also participates in the modulation of endocrine functions, such as steroid production [31] or insulin secretion [32]. The UT distribution highly resembles the UII distribution in cardiovascular, endocrine and also nervous tissues [33]. After a decade of research, in vivo data suggest that UII could participate rather in tissue remodelling processes during the course of the vascular disease [34] than in tonic vasculo-motor functions. This is also supported by the absence of modification of the vascular tone and only appearance of reduced metabolic syndrome and atherosclerotic lesions in UII knockout mice [35]. Recently, the impact of UT in the production of cytokines during immune response, the promotion of immune cell infiltration [36,37] as well as the pro-angiogenic functions [38] strongly suggest that the urotensinergic system exhibits chemokine properties. Indeed, our work previously demonstrated that pleotropic ability of UT not only involves coupling towards Gq/IP3 pathway, but also Gi/o/cAMP and G13 signalling, specific G protein effectors important for chemotaxic and mitogenic functions [33,39].

High-density UT has been reported in various tumour cell lines [40,41] as well as in human tumour extracts from lung [42], liver [43] and colon [44]. It has also been demonstrated that UII, at very low concentrations, stimulates cell proliferation and/or cell migration from human tumours, e.g., lung [45], colon [44], adrenal [40] cancers or high-grade gliomas [46,47]. Moreover, the UT receptor is present in peritumoral vascular smooth muscle cells, and is associated with macrophage infiltration [48] and vascular remodelling through endothelial cells [49]. These observations strongly suggest that the urotensinergic system plays a major role in induction and/or development of various solid tumours.

We considered that overexpression of UT in some solid tumours is a pejorative factor in tumorigenesis. It is hypothesised that novel radiolabelled peptides based on the urotensin II agonist, or on urantide, a biased ligand of UT, able to partially activate Gq but not receptor internalisation, should specifically bind UT expressed by solid tumour cells. Here we designed and evaluated two urotensinergic DOTA-conjugated analogues with the aim to allow imaging of solid tumours for diagnosis. Precisely, DOTA were coupled to human UII H_2_N-ETPDc-[CFWKYC]V-CO_2_H and urantide c-[Pen^5^, D-Trp^7^, Orn^8^]hUII4-11 via an aminohexanoic acid (Ahx) hydrocarbon linker to form DOTA-Ahx-hUII and DOTA-Ahx-urantide, respectively. These two radiolabelled analogues were developed because UII can bind UT in a quasi-irreversible manner [50] and induces internalisation while urantide, as a partial agonist, exhibits dual activity partially activating some G proteins without internalising properties (Scheme 1) [39]. These DOTA-hUII and DOTA-urantide exhibited similar binding affinities as native peptides and relayed calcium mobilisation in HEK293 cells expressing recombinant human UT. As a proof of concept for the use of UT as a functional target in solid tumours, we found that exposure of hUII stimulated the migration and proliferation of both human lung adenocarcinoma A549 and colorectal DLD-1 cell lines endogenously expressing UT. Using an in vivo approach, we finally evaluated the pharmacokinetic of the ^111^In-DOTA-hUII in immunocompetent mice, and the tumour targeting of both ^111^In-DOTA-hUII and ^111^In-DOTA-urantide in A549 and DLD-1 human tumours xenografted in nude mice.

## 2. Materials and Methods

### 2.1. Products and Reagents

Human urotensin-II (hUII) was obtained from Polypeptides Laboratories (Strasbourg, France) and urantide was from Peptide International (Louisville, KY, USA). The mouse monoclonal anti-MYC (sc-40), the rabbit monoclonal anti-UT (H-90) and the secondary antibody goat anti-mouse HRP (GAM-HRP) were purchased from SantaCruz Biotechnology (Paso Robles, USA). The mouse monoclonal anti-CD34 was obtained from Abcam and the secondary antibodies Alexia Fluor 488-conjugated donkey anti-mouse (DAM488) and Alexia Fluor 594-conjugated donkey anti-rabbit (DAR594) were from Invitrogen (Paris, France). Human UTS2R cDNA is inserted into the pCMV-MYC-N (UT-MYC) and pCMV-EGFP (UT-GFP) vectors. All constructs were previously verified by sequencing.

### 2.2. Synthesis of Human UII and Urantide Conjugated to DOTA

#### 2.2.1. Reagents and Materials

All chemicals, resins and solvents were used as received from the suppliers. Fmoc-protected amino acids, diisopropylethylamine (DIPEA), 2-(1*H*-7-Azabenzotriazol-1-yl)-1,1,3,3-tetramethyl uronium hexafluorophosphate methanaminium (HATU) and trifluoroacetic acid (TFA) were purchased from Chem-impex international (Wood Dale, IL, USA). Tri-tert-butyl 1,4,7,10-tetraazacyclo-dodecane-1,4,7,10-tetraacetate (tri-*t*Bu-DOTA-OH) was purchase from TCI (Tokyo, Japan). The 2-chlorotrityl chloride resin was obtained from Matrix Innovation (Québec, QC, Canada). Triisopropylsilane (TIPS) and ethanedithiol (EDT), iodine, formic acid and acetic acid, were obtained from Sigma-Aldrich corp (St-Louis, MO, USA). Dimethylformamide (DMF), isopropanol (IPA), methanol (MeOH), dichloromethane (DCM), diethyl ether and acetonitrile (ACN) were purchased from VWR (Ville Mont-Royal, QC, Canada). Optima grade Water was obtained from Fisher Scientific Canada (Toronto, Canada). Piperidine was obtained from A&C Chemicals (St-Laurent, QC, Canada). UPLC-MS analyses were performed with a Waters (Milford, MA, USA) AQUITY H-class – SQD2 mass detector and PDA el UV-visible detector on a BEH, C18, 1.7µm, 2.1 × 50 mm. Purifications were performed on a Waters preparative HPLC system consisting of injector 2707, pump 2535, and detector 2489, with an ACE C18 column 250 × 21.2 mm, 5 µm (Canadian Life Science, Peterborough, ON, Canada). For analytical UPLC, water and acetonitrile with 0.1% formic acid were used. For preparative HPLC, water plus 0.1% trifluoroacetic acid, and pure acetonitrile were used. Peptide syntheses were performed on Tribute automated peptide synthetiser from Protein Technologies (Tucson, AZ, USA) following manufacturer’s recommendations.

#### 2.2.2. Resin Preparation

The Fmoc-Valine-2-Chlorotityl resin was first prepared as follow. 2-chlorotrityl chloride resin (2-CTC, 1.0 g, 0.85 mmol/g) was shaken with a solution of amino acid (Fmoc-L-Val-OH, 0.30 mmol, 85 mg), diisopropylethylamine (DIPEA, 105 mL, 0.60 mmol) in 15 mL dichloromethane (DCM) for 1.5 h at room temperature. The 1 mL of 1/1 mixture of methanol and DIPEA was added to this solution and the resulting mixture was shaken for 10 min. After removing the excess reagents by filtration, the resin was washed consecutively with 5 mL DCM, 5 mL isopropanol, 5 mL DCM, 5 mL isopropanol, 5 mL DCM (3 min for each solvent) (standard washing). This result gave resin that has an approximate loading of 0.25 mmol/g.

#### 2.2.3. Synthesis on Tribute Synthetiser

Total of 200 mg Fmoc-Valine-2-chlorotrityl resin was charged in the reaction vessel of the Tribute automated peptide synthetiser. The typical cycle of synthesis correspond to the following description. Deprotection was performed using the smart deprotection feature of the synthetiser using 3 mL of the 20% piperidine in DMF solution. After the deprotection step, the resin was washed five times with DMF with 1.5 min of mechanical agitation for each wash. The coupling were performed using 5 eq of the amino acid, 5 eq of HATU and 10 eq of DIPEA in 4 mL of DMF for a period of 30 min and washed five times with DMF as previously mentioned. After the synthesis, Fmoc was removed manually (50% piperidine in DMF for 15 min, followed by standard washing) and the tri-*t*Bu-DOTA-OH (57 mg, 0.1 mmol, 2 eq) was coupled with HATU (38 mg, 0.1 mmol, 2 eq) and DIPEA (35 mL, 0.2 mmol, 4 eq) in 3 mL DMF for a period of 3 h. The final washing was performed and the completion of the coupling was checked using the Kaiser test.

Peptide was cleaved from resin using 5 mL cocktail trifluoroacetic acid (TFA)/triisopropylsilane (TIPS)/ethanedithiol (EDT)/water (92.5:2.5:2.5:2.5) and precipitate in 20 mL diethyl ether. After the supernatant was removed by centrifugation, peptide was resuspended in 4 mL of water-acetonitrile (3/1) and lyophilised. Then the peptides were resuspended in 4 mL of water-acetonitrile (3/1) and lyophilised a second time to remove all traces of ethandithiol. Peptide was dissolved in acetic acid 70% aqueous (1 mL/1 mg crude product) and disulphide bridge was made using a solution of 10% Iodine/MeOH (dropwise until persistent yellow colour observed). The solvent was removed by lyophilisation and peptide was purified by preparative HPLC. The pure fractions were combined and the final peptides were obtained after lyophilisation as white powders.

### 2.3. Radiolabeling and Stability

#### Optimisation of the Radiolabelling

For labelling with ^111^In, DOTA-hUII or DOTA-urantide was dissolved at 0.05 mM in 1 mL of isocitrate buffer (6 mM citric acid, 20 mM sodium citrate, 0.02 N chlorhydric acid, pH 4). ^111^InCl_3_ (30 MBq) was added to the vial and the solution was incubated at 100 °C for 20 min. The radiochemical purity of the radiolabelled peptides was determined by high performance liquid chromatography DIONEX U3000 HPLC system (HPLC, Thermofisher Ultimate 3000) equipped with a column (Kromasil^®^ KR100-5 C18 (250 × 4.6 mm, 5µm), and monitored with ultraviolet detector (wavelength: 254 nm) and with Gabi radioactivity detector from Elysia. Solvent A was water for HPLC and solvent B was acetonitrile. Data acquisition was performed by Chromeleon software.

DOTA-hUII or DOTA-urantide was dissolved at 0.05 mM in 1 mL of isocitrate buffer (6 mM citric acid, 20 mM sodium citrate, 0.02 N chlorhydric acid, pH 4). The prepared stock solution was stored at 4 °C protected from light.

A volume of 20 μL of each sample was injected on a C18 Kromasil^®^ column of 250 mm × 4.6 mm i.d., 5 μm particle size, (Nouryon), at 25 °C, with detection at λ = 254 nm coupled to radioactive detection. Mobile phase A was water for HPLC, while mobile phase B was acetonitrile. Separation of compounds was achieved using the following conditions: (i) Isocratic elution with 5% B from the injection time until 15 min, (ii) a gradient from 5% to 95% B over the next 10 min, (iii) an isocratic elution with 95% B for 5 min, another (iv) linear gradient from 95% to 5% B over 10 min and finally (v) isocratic elution with 5% B for 5 min. Equilibration was performed in a total analysis time of 60 min.

Samples for comparing the retention time between ^111^In-DOTA-hUII and DOTA-hUII reference standard were made by diluting, as well as ^111^In-DOTA-urantide and DOTA-urantide reference standard. For optimisation of the most efficient radiolabelling/chelation, the samples were prepared by mixing 1 mL of stock solution with various quantity of radioactivity ranging from 3.7 to 37 MBq of ^111^In. For in-vivo experiments, ^111^In-DOTA-hUII and ^111^In-DOTA-urantide stock solutions were prepared at a final concentration of 30 MBq/mL. All of the stock solutions were freshly prepared on the analysis day.

### 2.4. Serum Stability

Serum stability experiments were carried out on ^111^In-labeled hUII because of weekly availability of this isotope in the laboratory where the experiments were performed. To 1 mL of fresh human serum, previously incubated in a 5% CO2 environment at 37 °C, 0.6 nmol of the ^111^In-labeled hUII was added. The mixture was incubated in a 5% CO2, 37 °C and 100 μL aliquots (in triplicates) were removed at different time points and treated with 200 μL of EtOH to precipitate serum proteins. Samples were then centrifuged for 15 min at 2500*g* and after centrifugation, 50 μL of supernatant was counted, and also analysed by thin layer chromatography to determine the amount of intact peptide and its metabolites in the serum.

### 2.5. Cell Lines Culture and Transfections

Human lung adenocarcinoma A549 cell line was obtained from American Type Culture Collection (ATCC, CCL-185™). Human colorectal adenocarcinoma DLD-1 (ATCC, CCL-221™) cell line was provided by Dr L Grumolato (DC2N laboratory, Inserm U1239, Mont-Saint-Aignan, France) and human embryonic kidney HEK-293 (ATCC, CRL1573™) cell line was generously given by Dr Prézeau (IGF laboratory, Montpellier, France). All cell lines were routinely maintained according to the instructions from ATCC. More precisely, A549 and DLD-1 cells were cultured with RPMI 1640 media and HEK-293 cells were cultured with DMEM media, all supplemented with 1% sodium pyruvate (ThermoFisher Scientific, Montigny-Le-Bretonneux, France) and 10% foetal bovine serum (FBS, Lonza, Levallois-Perret, France). Transient transfections were performed using either Amaxa^®^ Cell Line Nucleofactor^®^ Kit V (Lonza, Levallois-Perret, France) or FuGene^®^ HD (Promega Corporation, Southampton, UK) according to the manufacturer’s protocol.

### 2.6. Binding Assay

Three micrograms of hUII in phosphate buffer (0.375 mM, pH 7.4) were labelled with 0.5 mCi Na^125^I (Amersham Biosciences) by the lactoperoxidase method as previously described [51]. Mono-iodinated [^125^I]hUII for the radioligand binding assays were purified by reversed-phase HPLC on an Adsorbosphere C_18_ column (0.46 × 25 cm, Alltech) using a linear gradient (25–65% over 40 min) of acetonitrile/trifluoro acetic acid (99.9:0.1, *v*/*v*) at a flow rate of 1 mL/min, and stored at 4 °C. The specific radioactivity of radioiodinated hUII was approximately 2000 Ci/mmol.

Cultured HEK-293 cells expressing UT-MYC were rinsed three times with PBS, dried under a cold air stream and stored at −80 °C until binding experiments. Frozen cells were washed twice with assay buffer (50 mM Tris buffer, 1 mM MnCl_2_ and 0.5% BSA) and incubated 2 h at 22 °C with ^125^I-hUII (0.2 nM) in the presence of graded concentrations of unlabelled hUII, DOTA-hUII, urantide or DOTA-urantide. At the end of the incubation, cells were washed three times with assay buffer, solubilised with 1% SDS and the radioactivity was counted in a gamma counter (LKB Wallac, Evry, France). Non-specific binding was determined by addition of 1 µM unlabelled hUII.

### 2.7. Calcium Assay

HEK-293 cells expressing UT-MYC were plated at 50,000 cells/well into clear-bottom, black, 96-well plates (Corning, Bagneaux-sur-Loing, France), previously coated with poly-L-ornithine (100 µg/mL, 1 h, 37 °C) and incubated overnight in culture medium. After 24 h, cells were rinsed twice with HBSS buffer (20 mM HEPES, 0.5 mM MgCl_2_, 2.6 mM CaCl_2_, 7.7 mM Na_2_CO_3_, 1.4 mM MgSO_4_, 5.3 mM KCl, 138 mM NaCl, 0.1% BSA, 5.5 mM glucose, 2.5 mM probenecid, pH 7.4) (Sigma Aldrich, Saint-Quentin-Fallavier, France) and loaded (40 min at 37 °C) with the calcium-sensitive dye fluo-4 AM (ThermoFisher Scientific, Montigny-Le-Bretonneux, France) containing pluronic acid (20% in DMSO). The effects of graded concentrations of hUII, DOTA-hUII, urantide and DOTA-urantide (10^−11^ to 10^−6^ M, each) on [Ca^2+^]_c_ were measured with a fluorometric imaging plate reader FlexStation III (Molecular Devices, Sunnyvale, CA). Data were expressed as the ratio of maximum response to the average control value.

### 2.8. Internalisation Assay

For ELISA experiments, HEK-293 cells expressing UT-MYC were plated at 10,000 cells/well into white, 96-well plates (Corning, Bagneaux-sur-Loing, France), previously coated with poly-L-ornithine (100 µg/mL, 1 h, 37 °C) and incubated overnight in culture medium. Cells were treated for 1 h with increasing concentrations of hUII, DOTA-hUII (10^−13^ to 10^−6^ M, each), urantide or DOTA-urantide (10^−10^ to 10^−6^ M, each), washed with PBS and fixed with paraformaldehyde (4%, 10 min, 20 °C). Cells were permeabilised or not with Triton X-100 (0.05%, 5 min). Non-specific binding sites were blocked with FBS (10%) and normal donkey serum (NDS, 2%, 1 h). Cells were then incubated with anti-MYC antibody (1 h, 20 °C) followed by incubation with GAM-HRP (1 h, 20 °C). After washes, chemoluminescence was detected with an ELISA kit (ThermoFisher Scientific, Montigny-Le-Bretonneux, France) and the InfinitePro200 plate reader (TECAN FRANCE, Lyon). Data are expressed as the ratio between intensity measured on the permeabilised cells and that measured on the non-permeabilised cells, compared to the average control value.

For cytochemistry experiments, HEK-293 cells expressing UT-GFP were seeded at 70,000 cells/well into transparent, 24-well plates (Corning, Bagneaux-sur-Loing, France), previously coated with human fibronectin (25 µg/mL, overnight, 4 °C) and incubated overnight in culture medium. Cells were treated for 1 h with hUII, urantide, DOTA-hUII or DOTA-urantide (10^−7^ M, each). Cells were rinsed with PBS, fixed with paraformaldehyde (4%, 10 min) and counterstained with DAPI (1 µg/mL, 5 min) to label nuclei. Image analysis was performed by confocal microscopy (Leica TCS SP8 confocal laser scanning microscope).

### 2.9. Cell Migration Assay

DLD-1 and A549 cells were seeded (50,000 cells) on transwell filters with 8-µm pores previously coated with human fibronectin (25 µg/mL, 4 °C, overnight) in serum-free medium on the upper chamber. A chemotaxic diffusion was generated through the microporous membrane by filling the lower chamber with hUII (10^−9^ M or 10^−7^ M). After 24 h incubation period at 37 °C, filters were rinsed with PBS and non-migrating cells were removed from the upper surface by using a cotton-swab. Migrating cells were fixed in successive baths of methanol (50, 70, 90 and 100%), stained with eosin and haematoxylin and filters were mounted on glass slides with Mowiol. Random phase contrast images of filters were then recorded, using a digital camera connected to a Nikon microscope (Eclipse TS-100) and image analysis was performed by using *Image J*’s Cell Counter. The data were represented as the ratio of treatment chemoattraction normalised to control.

### 2.10. Animal Models

All procedures were performed in accordance with the French Ethical Committee as well as the guidelines of European Parliament directive 2010/63/EU and the Council for the Protection of Animals Used for Scientific Purposes. This project was approved by the ‘Comité d’Ethique NOrmandie en Matière d’EXpérimentation Animale’ CENOMEXA under the National Committee on Animal Experimentation, and received the following number 2017031615007139. Animal manipulations were carried out under the supervision of an authorised investigator (H. Castel; authorisation no. 76.98 and microsurgery (2014); P. Gandolfo authorisation no. 76.A.25 from the Ministère de l’Alimentation, de l’Agriculture et de la Pêche). The criterion used here as defined in the guidelines on radioligand development is an early preclinical validation of a radiotracer. Our study is a proof of principle of the evaluation of the in vivo biodistribution, blood clearance and optimally, the tumour capture of novel radiolabelled urotensinergic ligands. Thus, a minimum number of animals were used to find a ‘true effect’ of the intravenous injection of radiolabelled ligands in mice. This is in agreement with the publication by Vanhove et al. dealing that the number of animals can be rather small (around 5 per group) to reach significance or an accurate information if efforts are made to reduce intra-animal variability [52].

For biodistribution studies, 8-week-old female C57BL6/6 (René Janvier Laboratories) were used (n = 4 per group). For tumour capture studies, 8-old-week female Swiss (*nu*/*nu*) mice (Charles River Laboratories) were used (n = 5 per group). Mice were acclimated and housed in sterile cages in groups of 5, in a temperature-controlled room with a 12-h light/12-h dark schedule and fed with autoclaved food and water *ad libitum*.

### 2.11. Immunochemistry

A549 or DLD-1 tumour cryostat sections (10 µm in size) were processed in a Leica CM1950 Cryostat, mounted directly on slides and then fixed in 4% paraformaldehyde (10 min, 20 °C), permeabilised with Triton X-100 (0.05%, 5 min) and non-specific sites were blocked with FBS and NDS (10% and 2%, respectively, 1 h). Slices were incubated overnight at 4 °C with primary antibodies (anti-UT and anti-CD34) followed by incubation for 2 h at 20 °C with appropriate secondary antibodies (DAR594 and DAM488, respectively). Slices were counterstained with DAPI (1 µg/mL, 10 min) to label nuclei, and imaged by confocal microscopy (Leica TCS SP8 confocal laser scanning microscope).

### 2.12. In Vivo Assay

For biodistribution studies, C57BL6/6 wild-type mice were injected with ^111^In-DOTA-hUII (3 MBq, 100 µL) into the tail vein. Mice were sacrificed at 4, 24, 48 and 72 h post-injection and blood and organs were collected. All samples were analysed with a gamma counter and the data presented were calculated as the percent injected dose per gram of tissue (% ID/g). Data were expressed as mean ± SEM obtained from four animals per kinetic time.

For tumour targeting studies, inocula (100 µL) containing a suspension of 5 × 10^6^ freshly harvested A549 or DLD-1 cells in phosphate buffered-saline were injected subcutaneously into the right shoulder blade in Swiss (nu/nu) mice (n = 5 in each group). When the tumour size reached 250 mm^3^ (approximatively 2-weeks), 2 MBq of ^111^In-DOTA-hUII or ^111^In-DOTA-urantide were injected into the tail vein. After 24 h, animals were sacrificed, and tumours were collected, directly frozen in isopentane (−50 °C) and stored at −80 °C.

### 2.13. Statistical Analyses

Data were expressed as mean ± SEM and analysed with GraphPad Prism (version 7, GraphPad Software, Inc. San Diego, CA, USA). Student *t* test was used for parametric comparisons, Mann-Whitney *U* test was used for non-parametric comparisons, and multivariate analysis were done with ANOVA one-way test. All reported *p* values were two-sided and considered to be statistically significant at *p* < 0.05.

## 3. Results and Discussion

### 3.1. Synthesis and Radiolabelling of DOTA-hUII and DOTA-Urantide

GPCRs play a major role in the initiation and progression of cancers. Several of them, such as angiotensin-1 (AT1), endothelin-B (ETB) or CXCR4 receptors, involved in a wide range of biological mechanisms, participate in the modulation of proliferation/migration and/or angiogenesis, three fundamental processes involved in tumorigenesis [1,53,54]. Some of these GPCRs are over-expressed in tumour cells, constituting interesting targets for the diagnosis and/or treatment of solid tumours. For example, somatostatinergic radiolabelled analogues have been developed to image neuroendocrine tumours, which contain high density of sst_2_ receptors, therefore, the discovery of new GPCR over-expressed in tumours, is a promising way to develop new radioligands [3,6,55]. The neuropeptide UII also participates in critical tumorigenic mechanisms, such as cell migration and/or proliferation [36,46,56] and tumour angiogenesis [57]. The urotensinergic receptor UT is present at the plasma membrane of tumoral cells from lung [42], breast [41], colon [58] or prostate cancer [59]. It has long been accepted that the internalisation of an agonist ligand-receptor complex justifies their use in diagnosing cancer, likely because of the accumulation of radioligand in the tumour cells [60,61]. However, other studies demonstrated that two somatostatinergic antagonists ^111^In-DOTA-sst2-ANT and ^111^In-DOTA-sst3-ODN- are capable of accumulating in vivo in HEK-293 cells expressing sst2 and sst3 respectively and transplanted into mice [62]. This interesting usefulness of antagonist is likely due to high accessibility of binding sites [63] associated with slow dissociation from the receptor. In our study, it thus motivated the design of the radiolabelled DOTA-hUII agonist and DOTA-urantide biased/antagonist ligand to find the best diagnosis compound targeting UT expressed in some solid tumours.

DOTA-Ahx-hUII and DOTA-Ahx-urantide analogues were synthesised by solid-phase Fmoc synthesis and purified by RP HPLC. Their sequences were confirmed by ESI mass spectrometry to finally display greater than 95% purity after purification (Figure 1). DOTA chelating agent was chosen because of the better thermodynamic stability of the final complex compared to DTPA complexes [13,64]. In addition, DOTA can also incorporate a wide range of radioisotopes used for diagnosis, such as ^111^In and ^68^Ga [64,65], or therapy, such as ^90^Y and ^177^Lu [66,67] and, more recently tested in preclinical research, ^213^Bi [68]. Indeed, the large number of radioisotopes that can be coupled to DOTA makes its exploitation in theranostics possible [69,70,71]. Since the addition of a DOTA in the biologically active cyclic hexapeptide could prevent binding to the receptor, the N-terminal sequences of UII and urantide are the optimal sites for chelator conjugation.

In addition, the linker Ahx, which acts as a spacer between the chelator and the urotensinergic analogues, avoids steric hindrance by moving the chelator away from the active site of the peptides, thus not altering the site of ligand-receptor interaction [72,73]. ^111^In-DOTA-hUII and ^111^In-DOTA-urantide were finally prepared as 80 to 95% radiolabelled peptides in isocitrate buffer at 100 °C (Figure 1).

Figure 2 shows that radiolabelling of DOTA-hUII or DOTA-urantide with ^111^In was achieved by 20 min incubation with ^111^InCl_3_ in isocitrate buffer (pH 4) at 100 °C, while 95 °C failed to achieve 50% labelling efficacy of both DOTA-peptide analogues (Figure 2A). For both analogues, the incorporation in the DOTA chelator was >99%, as demonstrated by radioanalytical HPLC (Figure 2B). This high temperature to reach labelling efficiency can be explained by the existence of a barrier of potential energy to overcome in order to achieve DOTA-Indium complexation. It was often experimentally observed that heating must be long enough to achieve this complexation but to accelerate the reaction; high temperatures have often been used during the complexation [74]. Accordingly, Breeman et al. has shown that the complexation of indium-DOTA is much better at 100 °C and DOTA-OC was successfully obtained in complex with ^111^In at 100 °C [64].

The specificity of the analytical method was evaluated by identifying the retention times obtained from DOTA-hUII, DOTA-urantide, ^111^In-DOTA-hUII, ^111^In-DOTA-urantide and ^111^In-Citrate. The retention times of DOTA-hUII and DOTA-urantide are 22.14 and 22.03 min, respectively (Figure 2C). These data show that the retention times of precursors (DOTA-hUII or DOTA-urantide) interfere with indium radiochemicals, indicating that the specific radioactivity is dependent from the quantity of cold reactant.

The radiochemical stabilities of the ^111^In-DOTA–hUII analogue was assessed by incubation in human serum at 37 °C. After 1 and 3 h spent in human plasma, more than 80 and 70% of intact radiopeptide, respectively, were found (Table 1), providing a comfortable time window for in vitro and in vivo experiments [8,75].

These results are broadly in agreement with the literature, which indicates that peptides containing a Met, Ser, Ala, Thr, Gly or Val residue in the N-terminal position, such as DOTA-UII or even DOTA-urantide, have a longer half-life than other compounds [76]. Together, two DOTA urotensinergic analogues (DOTA-hUII and DOTA-urantide) have been synthesised.

### 3.2. Recognition and Activation of UT Receptor by DOTA-hUII and DOTA-urantide

The addition of DOTA in a peptide sequence may alter affinity and binding to the receptor and receptor-induced cellular response, as previously described with NPY [77]. Thus, to verify that conjugation of DOTA in hUII and urantide did not alter their affinity and selectivity for UT, we first evaluated the ability of DOTA-hUII and DOTA-urantide to interact with UT. Competition binding studies performed with ^125^I-hUII on HEK-293 cells expressing human UT showed that conjugation of DOTA on hUII or urantide did not significantly modify the affinity of the peptide (IC50 (nM):hUII, 6.01 ± 1.47; DOTA-hUII, 9.13 ± 2.06; urantide, 28.98 ± 11.19; DOTA-urantide, 47.63 ± 11.23) (Figure 3A,C). UT receptor is classically coupled to the phospholipase C/IP_3_/Ca^2+^ pathway [27,51].

We thus compared the ability of hUII, urantide and their DOTA analogues to mobilise Ca^2+^ in HEK-293 cells expressing human UT. Exposure of cells to graded concentrations of DOTA-hUII or DOTA-urantide (10^−11^ to 10^−6^ M, each) induced a transient and dose-dependent increase in [Ca^2+^]_c_ (EC_50_ (nM):19.44 ± 2.47; 32.14 ± 6.94, respectively) with a potency and efficacy close to those obtained with hUII and urantide (EC_50_ (nM):8.05 ± 0.67; 50.47 ± 16.12, respectively) (Figure 3B,C). These results demonstrated that conjugation of a DOTA with Ahx linker at the N–terminal sequences of hUII or urantide did not impair the abilities to bind UT, and to trigger a UT-induced calcium response. This constitutes a first advantage as diagnosis radioligand. In addition, the existence of a single urotensinergic receptor known to date constitutes a singularity in the field of neuropeptides, while octreotide can interact preferentially with sst2, sst3 but also sst5. This characteristic of the urotensinergic system is therefore a significant advantage since it can be used to develop optimised analogues for tumours overexpressing the UT receptor. Moreover, the good labelling and affinity results obtained with DOTA-urantide, a biased ligand, appears very promising since this type or radioligand should in theory exhibit high affinity, stability and lipophilicity while blocking UT oncogenic signalling cascades and avoiding receptor internalisation.

### 3.3. Internalisation of UT Evoked by Urotensinergic DOTA Analogues

Development of GPCR DOTA-radioligands to image solid tumours requires internalisation of the receptor/complex, allowing radioactivity accumulation in tumour cells [77]. However, several in vivo studies have shown that an absence or low internalisation is sometimes correlated with very high tumour retention [62,63,78].

The ability of urotensinergic DOTA analogues to provoke UT internalisation was first evaluated by using an ELISA assay on HEK-293 cells transfected with cDNA encoding the human tagged with c-myc. Application for 1 h of hUII (10^−13^ to 10^−6^ M) provoked a dose-dependent increase in internalisation rate of UT, starting from picomolar concentrations and reaching a plateau between 10^−9^ and 10^−8^ M (Figure 4A). As expected, DOTA-hUII exerted similar effects, with very close potency and efficacy. In contrast, both urantide and its DOTA analogue failed to provoke UT internalisation, even at higher concentrations (Figure 4A). These results were confirmed by confocal microscopy of HEK-293 cells transiently expressing human UT-GFP. In control conditions, UT is mostly located at the plasma membrane (Figure 4B). Application of hUII or DOTA-hUII (10^−7^ M, each) for 1 h promoted a strong accumulation of UT in cytosolic puncta into cells, while urantide and DOTA-urantide were both ineffective (Figure 4B). These data are in good agreement with our previous original study which established that urantide exhibit biased signalling between β-arrestin and G proteins, and between Gq, Gi/o and G13-protein subtypes, likely maintaining UT at the cell surface and preventing UII binding [39].

### 3.4. UT Expression and Function in DLD-1 and A549 Tumour Cell Lines

Currently, Indium–111 is used for imaging tumour in patients or animals, mainly through its interesting characteristics, e.g., compatibility with many chelator including DOTA, emission of γ rays detectable with a gamma camera, relatively long half-life compared with other radioisotopes, thus offering a reasonable time to perform in vivo experiments requiring late imaging [2,8]. Here we hypothesised that the two urotensinergic radiolabelled analogues may constitute lead compounds for solid tumour diagnosis. Previous studies have reported the presence of UT on several human cell lines, including DLD-1 or A549 [45], originally isolated from colorectal adenocarcinoma and lung adenocarcinoma, respectively. By a Western blot analysis, we first verified the presence of UT in both cell lines, and we observed a major band at the expected molecular mass (≈ 65 kDa) (Figure 5A, left). To test whether the expression of UT is also observed in the tumour microenvironment, immunohistochemical studies were then performed from xenografts obtained after heterotopic injection of DLD-1 or A549 in nude mice. The receptor was mostly located within and in the periphery of angiogenic components expressing CD34 (Figure 5A, right).

The ability of hUII to promote migration and/or proliferation in these cell lines was demonstrated by means of the modified Boyden chamber assay. As illustrated in Figure 5B, hUII stimulated chemotactic migration in DLD-1 and A549 cells, with a more potent activity observed on DLD-1 cells. Furthermore, hUII was able to increase the proliferation rate of both tumour cell lines only observable after 48 h period incubation, at low and high hUII concentrations (Figure 5C). These data are in a good agreement with studies showing that UT and/or UII are over-expressed in a wide variety of solid tumours [41,42,43,44,58] and participate in tumorigenic mechanisms such as cell migration and/or proliferation. These data have been reported for cancers of prostate [59], colon [44], liver [79] and lung [48] cancers.

### 3.5. Biodistribution of ^111^In-DOTA-hUII and ^111^In-DOTA-Urantide

The in vivo experiments initially consisted of measuring the biodistribution of the radioligand in the small animal, thus making it possible to assess its uptake in the organism, particularly in critical organs such as the liver and kidneys [80,81]. The biodistribution of ^111^In-DOTA-hUII was first tested in 8-old-weeks female C57B/L6 mice, after samplings of many organs at 4, 24, 48 and 72 h post-injection and tested by means of a γ counter (Figure 6A). The results were calculated as mean %ID/g value ± SD, and are summarised in Figure 6A. ^111^In-DOTA-hUII displayed a rapid clearance from blood and background tissues predominantly via the kidney. In fact, more than 90% of radioactive DOTA-peptide was cleared from animals over 4 h post-injection. Studies of UT distribution in mice, which are less numerous than those conducted in humans or rats, show significant expression of the receptor in the heart, pancreas and kidneys [33]. However, analysis of ^111^In-DOTA-hUII uptake revealed a weak signal in these organs. Furthermore, the estimated amount of radioactivity found in blood, the digestive system (liver, intestines, etc.) and other organs is two to three times less than that found for ^111^In-DOTA-ligands tested by other teams, suggesting that ^111^In-DOTA-hUII would be rapidly filtered and eliminated in the urine [55,82]. At this stage, this elimination could limit the use of ^111^In DOTA-hUII, because the ligand would fail to reach and access to its target, thus compromising the use of the radioligand to detect tumour foci.

In order to test the potential targeting of solid tumours expressing UT, heterotopic xenografts were obtained by injection of A549 cells (from human lung adenocarcinoma) or DLD-1 cells (from human colorectal adenocarcinoma) in nude mice. When tumour size reached 250 mm^3^ (variable kinetic, see Figure 6B), ^111^In-DOTA-hUII or ^111^In-DOTA-urantide were injected into the tail vein as shown in the protocol and tumour, muscle, kidney and liver were collected 24 h post-injection to be counted. We found reduced kidney uptake of radiolabelled-DOTA-urantide compared with DOTA-hUII, likely suggesting better distribution of the UT biased DOTA-urantide but similar tumour/muscle ratio, e.g. ^111^In-DOTA-hUII, 3.33 ± 0.96 and 2.85 ± 0.66; ^111^In-DOTA-urantide and 2.47± 0.15 and 2.08± 0.45 for DLD-1 and A549, respectively (Figure 6B). Although internalisation (and accumulation after 24 h) of the radioligand within receptor overexpressing tumours could be expected at least for ^111^In-DOTA-hUII, these results ultimately corroborate those of ^111^In-DOTA-hUII biodistribution in C57BL/6 mice. Despite fast clearance, studies conducted with 99mTc-TP3654 [83], ^111^In-DTPA-octreotide [84] or 64Cu-DOTA-TATE [85] had allowed visualisation of the targeted tumours. But our current ^111^In-DOTA-hUII analogues do not satisfy the requirements for human solid tumour diagnosis and treatment. First is to increase the number of carbons between the ligand and the chelator to improve the pharmacodynamic properties of the analogues, such as (i) the possible backbone modification for plasma stability and modifications of single amino acids for reduction of kidney uptake [86], or (ii) the addition of macromolecules such as glucose, folate, IgG or recognition molecules to albumin [18,87].

The second is to combine the analogues with other ligands, such as RGD, to generate bi-functional radioligands targeting the pro-angiogenic integrin αvβ3 generally overexpressed by endothelial cells in tumour foci. These different strategies could therefore be explored to optimise urotensinergic DOTA-peptides in order to make them operational in the detection of tumours overexpressing the UT receptor. In 2015, Reynolds et al. synthesised RGD-Glu-DO3A-Ahx-RM2 capable of binding to gastrin-releasing peptide receptor and the integrin αvβ3 involved in tumour angiogenesis [73,88,89]. Administered to mice with PC-3 cell xenografts (from prostate adenocarcinoma), this radioligand is characterised by a tumour/muscle ratio close to 50, limited uptake in other organs and rapid renal clearance. The involvement of the urotensinergic system in vascular remodelling through endothelial cells, its over-expression in tumour cells and its ability to stimulate macrophage infiltration within tumours [90] suggest that the development of RGD-DOTA-peptide urotensinergic analogues would therefore also be a promising strategy.

## 4. Conclusions

In conclusion, we have developed for the first time two DOTA-urotensinergic peptides, one focused on the agonist status of hUII and the other on the biased/antagonist property of urantide by using the ^111^Indium radiobiological tracer. Importantly, the conjugation of the DOTA chelator in the N-terminal extremity of hUII or urantide did not alter UT binding affinities and activation. However, in vivo intravenous injections in mice show rapid uptake in kidney and rapid renal clearance while low access to lung and colorectal xenografted tumours functionally expressing UT.

Some interesting aspects suggest that the receptor UT of UII is a target of choice for diagnosis imaging and therapeutic strategies of various solid cancers, including glioblastoma. First, UT and the autocrine/paracrine actions of UII are pejorative factors in tumour evolution. Second, the existence of a single urotensinergic receptor known to date constitutes a singularity in the field of neuropeptides that should provide a significant advantage for in vivo imaging and tumours targeting while reducing side organ toxicities. The fast clearance must be overcome by using strategies likely employed to increase the molecular weight of the radioligand (via adding polymers between the chelator and the ligand) or to prolong the radioligand half-life by conjugation with a carbon chain fatty acid residue allowing binding to albumin. However, considering that UT can be expressed by tumour cells and angiogenic components within tumour bulk, we think that an optimum future strategy should be based on the design of RGD-DOTA-Ahx-hUII/urantide bi-functional molecules capable of recognising multiple tumour cells and key integrins as αvβ3 associated with angiogenesis. If combined with different radiotracers such as ^68^Ga, imaging of solid tumours including glioblastoma should be highly efficient.

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
