# Peer review of "Development of Novel 111-In-Labelled DOTA Urotensin II Analogues for Targeting the UT Receptor Overexpressed in Solid Tumours"

_biomolecules, 2020, doi:10.3390/biom10030471_

Round 1

Reviewer 1 Report

This manuscript titled “Development of novel 111-In-labeled DOTA urotensin 2 II analogues for targeting the UT receptor 3 overexpressed in solid tumours” by Poret et al described synthesis and characterization of molecular imaging probe for UT. The reviewer believes this is the first attempt to study radiolabeled DOTA-peptide ligands for UT, even though 18F labeled UTII was prepared previously.

While in vivo imaging experiment failed due to the fast clearance according to the authors, the study is well designed and worth publication. There are a few minor points that need attention, though.

1) In Scheme 1, crosslinking between cysteines or penicillamine chould be drawn between the side chain. Abbreviation “Pen” is hard to recognize the presence of thiol in the side chain.

2) Line 230, Page 7, Cos2 should be corrected

3) Line 336, Page 9, In vivo should be italic

4) Line 341, Page 9, 5.106 should be corrected.

5) Line 384, Page 11, C-terminal sequence is not correct. DOTA is conjugated to the N-terminus of the peptide.

6) Figure 1, DOTA-Ahx-Asp-[Pen-Phe-DTrp-Orn-Tyr-Cys]-Val-OH and DOTA-Ahx-Glu-Thr-Pro-Asp-[Cys-Phe-Trp-Lys-Tyr-Cys]-Val-OH should be swapped to match the structures below.

7) Line 402, Page 12, the authors could elaborate why the labeling efficacy is so different between 95 oC reaction and 100 oC reaction.

8) Line 424, Page 13, N-terminal position should be C-terminal position. Val in the peptides is C-terminal residue.

9) Line 442 and 444, Page 13, superscripts should be shown in 125I and 10-11 and 10-6M

10) Line 452, Page 14, at the C-terminal should be at the N-terminal

11) Line 560, Page 18, the authors suggested the longer carbon linker that will affect the binding affinity by increasing lipophilicity as shown in the reference (90). However, the DOTA-UTII peptide already showed very good binding affinity and the reviewer do not agree with the author’s argument. Furthermore, “ several potential alternatives to overcome these difficulties have been already tested.” should be removed. They were not tested on these DOTA-peptide clearance issue.

12) Line 562, Page 18, PEG is in conflict with carbon liner in Line 560. PEG linkers are considered hydrophilic.

Author Response

Reviewer 1

This manuscript titled “Development of novel 111-In-labeled DOTA urotensin 2 II analogues for targeting the UT receptor 3 overexpressed in solid tumours” by Poret et al described synthesis and characterization of molecular imaging probe for UT. The reviewer believes this is the first attempt to study radiolabeled DOTA-peptide ligands for UT, even though 18F labeled UTII was prepared previously.

While in vivo imaging experiment failed due to the fast clearance according to the authors, the study is well designed and worth publication. There are a few minor points that need attention, though.

Answer to reviewer 1

I would like to thank the reviewer for his Positive appreciation and his constructive comments and recommanded corrections

1) In Scheme 1, crosslinking between cysteines or penicillamine chould be drawn between the side chain. Abbreviation “Pen” is hard to recognize the presence of thiol in the side chain.

In order to clarify this point, and to highlight the presence of thiol in the side chain, another type of illustration of the two peptides sequence are proposed in the revised Scheme 1. See new scheme 1 in the revised manuscript. P4, line 135.

2) Line 230, Page 7, Cos2 should be corrected:

This typo is now corrected, p7, Line 242.

3) Line 336, Page 9, In vivo should be italic:

The typo was corrected, p10, line 361.

4) Line 341, Page 9, 5.106 should be corrected.

The Typo was corrected,.

5) Line 384, Page 11, C-terminal sequence is not correct. DOTA is conjugated to the N-terminus of the peptide.

This mistake was corrected, thank you, p11, line 411.

6) Figure 1, DOTA-Ahx-Asp-[Pen-Phe-DTrp-Orn-Tyr-Cys]-Val-OH and DOTA-Ahx-Glu-Thr-Pro-Asp-[Cys-Phe-Trp-Lys-Tyr-Cys]-Val-OH should be swapped to match the structures below.

We apologize for this mistake, this was corrected in the revised manuscript, p12, line 426.

7) Line 402, Page 12, the authors could elaborate why the labeling efficacy is so different between 95 oC reaction and 100 oC reaction.

This high temperature to reach labeling efficiency can be explained by the existence of a barrier of potential energy to overcome, in order to achieve DOTA-Indium complexation. It was often experimentally observed that heating must be long enough to achieve this complexation but to accelerate the reaction; high temperatures have often been used during the complexation (Léon-Rodriguez et al., Bioconj chem 2008, 19 : 391-402). Accordingly, DOTA-TOC, a close analog of urantide, was successfully obtained in complex with 111In at 100°C (Léon-Rodriguez et al., Bioconj chem 2008, 19 : 391-402) and Breeman has shown that the complexation of Indium-DOTA is much better at 100°C (Breeman et al., EJNMMI 2003). This point was discussed in the revised manuscript p13, lines 442-447, and the reference by Leon-Rodriguez was cited.

8) Line 424, Page 13, N-terminal position should be C-terminal position. Val in the peptides is C-terminal residue.

It was here proposed that complexation/stability of DOTA-peptides can be favored by N-terminal amino acids such as Met, Ser, Ala, Thr, Gly or Val. Thus, the amino acid Glu in the N-terminus of hUII and Asp in the N-terminus of urantide thus would lead to better coupling to DOTA.

9) Line 442 and 444, Page 13, superscripts should be shown in 125I and 10-11 and 10-6M

The typos were now corrected, p14, line 484.

10) Line 452, Page 14, at the C-terminal should be at the N-terminal

We apologize for this typo, it was corrected, p15, line 496.

11) Line 560, Page 18, the authors suggested the longer carbon linker that will affect the binding affinity by increasing lipophilicity as shown in the reference (90). However, the DOTA-UTII peptide already showed very good binding affinity and the reviewer do not agree with the author’s argument. Furthermore, “ several potential alternatives to overcome these difficulties have been already tested.” should be removed. They were not tested on these DOTA-peptide clearance issue.

The reviewer is right, thus we removed the reference from Guo et al., 2013 [90] and focused on the reference by Valverde to evoke the possible backbone modification for plasma stability and modification of a single amino acid for reduction of kidney uptake. Accordingly, numbering of the references was re-adjusted in the revised manuscript, also taking into account the comments of reviewer 2.

12) Line 562, Page 18, PEG is in conflict with carbon liner in Line 560. PEG linkers are considered hydrophilic.

In agreement with the important remark of the reviewer, we do not consider PEG linker except to increase the molecular weight of the ligand (in conclusion section). Thus we removed this example in the revised manuscript. p19, line 602.

I would like to thank you for your time in consideration,

Dr Hélène Castel

Reviewer 2 Report

The manuscript entitled: “Development of novel 111-In-labeled DOTA urotensin 2 II analogues for targeting the UT receptor overexpressed in solid tumours” is intersting and properly assessed. It fits scope and aims of the Journal adding information to the area of interest. Abstract section should be summarised focusing on the end points of the manuscript and experimental. The experimental part is clear and properly described. Results are justified. Regarding the study on animals: have the minumum number of animals for the experiments been used? Please specify in the ethycal committee comments section the criterion used for in vivo assays. The Conclusion section should be exploited giving also the Authors point of view in more detail regarding future research in the explored field or research. Some of the oldest References should avoided: please check.

Author Response

Reviewer 2:

We thank the reviewer for his favorable opinion and the constructive recommandations to improve the quality of the manuscript.

1)Abstract section should be summarised focusing on the end points of the manuscript and experimental.

As importantly recommended by the reviewer, the abstract was rewritten to follow experimental end points and to highlight the main objective. p1, lines 21-37 and p2, lines 38-58.

2) Regarding the study on animals: have the minumum number of animals for the experiments been used? Please specify in the ethycal committee comments section the criterion used for in vivo assays.

We agree with the reviewer that the number of animals per group would appear rather small. We conducted our in viv oresearch by following the EU Directive 2010/63/EU of the European Parliament and of the Council of 22 September 2010 on the protection of animals used for scientific purposes (OJ L 276, 20.10.2010) and The ARRIVE Guidelines—Animal Research: Reporting In Vivo Experiment. Our in vivo protocol was authorized by the local and the French ethic committees.

The number of animals was reduced as much as possible (n = 5) per group and time point for all in vivo experiments following the 3R rules (Turner PV, et al. Roles of the international council for laboratory animal science (ICLAS) and international association of colleges of laboratory animal medicine (IACLAM) in the global organization and support of 3Rs advances in laboratory animal science. J Am Assoc Lab Anim Sci. 2015;55:174–180. The criterion used here as defined in guidelines for radioligand development is “an early preclinical validation of a radiotracer”. Our study is a proof of principle of the evaluation of the in vivo biodistribution, blood clearance and optimally, the tumor capture of novel radiolabeled urotensinergic ligands. Thus, a minimum number of animals should be required to find a “true distribution effect” of the intravenous injection of a radiolabeled ligand in mice. This is in agreement with the publication by Vanhove et al., EJNMMI Physics (2015) 2:31, dealing that the number of animals can be rather small (around 5 per group) to reach significance or accurate information (detection of tracer versus no radioactivity) while reducing intra-animal variability (around 10%).

This precise information was added in the material and method section and the reference by Vanhove et al., 2005, was now cited. P9, lines 323-342.

3) The Conclusion section should be exploited giving also the Authors point of view in more detail regarding future research in the explored field or research.

We agree with the reviewer that it should be a plus, and we provided our opinion in the revised manuscript, proposing that bifunctional ligands targeting angiogenic and tumoral components should be a promising strategy for urotensinergic ligands, applicable to glioblastoma. p20, lines 623-648, p21, lines 649-650.

4) Some of the oldest References should avoided: please check

As recommanded we removed the non-essential oldest references, and references were renumbered

I would like to thank you for your time in consideration,

Best Regards,

Dr Hélène Castel